# No Effect of Intermittent Palm or Sole Cooling on Acute Training Volume during Resistance Exercise in Physically Active Adults: A Summary of Protocols

**DOI:** 10.3390/sports12100281

**Published:** 2024-10-16

**Authors:** Rouven Kenville, Martina Clauß, Aleksander Arup, Patrick Ragert, Tom Maudrich

**Affiliations:** 1Department of Movement Neuroscience, Faculty of Sports Science, Leipzig University, 04109 Leipzig, Germany; 2Department of Neurology, Max Planck Institute for Human Cognitive and Brain Sciences, 04103 Leipzig, Germany

**Keywords:** intermittent cooling, palm cooling, resistance exercise, sole cooling, training volume

## Abstract

Intermittent palm (PC) and sole cooling (SC) are proposed ergogenic methods for enhancing exercise performance during high-intensity and fatiguing conditions. However, findings in the literature regarding its positive effect remain inconclusive. This study aimed at investigating the effects of intermittent PC and SC compared to no cooling (NC) on acute training volume during resistance exercise, particularly focusing on the total number of repetitions (TR) performed. Three separate randomized crossover protocols, incorporating commonly practiced resistance exercises (Protocol 1: pullups; Protocol 2: pushups; Protocol 3: leg extensions), were conducted, enrolling healthy, physically active adults (overall sample: n = 41 (12 female), age: 23.9 ± 4.0 years (mean ± SD), height: 174.4 ± 9.5 cm, body mass: 69.3 ± 12.4 kg). During Protocol 3, tympanic temperature (TT), rate of perceived exertion (RPE), and electromyography (EMG) of quadriceps muscles were additionally assessed for SC. PC resulted in less TR compared to NC in Protocol 1 (*p* < 0.001). Protocol 2 and 3 did not reveal significant ergogenic benefits of PC or SC compared to NC (*p* > 0.05). Furthermore, SC had no effect on TT, RPE, or EMG amplitudes (all *p* > 0.05). The inconsistent findings suggest that intermittent PC and SC might have limited effectiveness in enhancing training volume during resistance exercise in physically active adults. Future research should examine various resistance training protocols under controlled conditions, and incorporate comprehensive physiological measurements to elucidate the potential benefits and mechanisms of intermittent cooling in resistance exercise contexts.

## 1. Introduction

Resistance exercise is known to improve overall health and physical performance [1], particularly in the domains of strength, hypertrophy, and muscular endurance. Advancing physical performance through resistance training commonly involves progressive overload, a principle that entails gradually increasing overall training stimulus. This in turn drives continuous adaptations and performance improvement [2]. Since progressive overloading requires a continuous, periodic increase in either load, repetitions per set, or sets (training volume), modern ergogenic strategies increasingly focus on enhancing training volume. One such method that has gained prominence in recent years is inter-set intermittent cooling of body parts, e.g., hands and feet during resistance training [3]. This technique is grounded in the premise that skeletal muscles operate most efficiently within a specific temperature range [4]. During prolonged resistance exercise, muscle temperature increases [5], which initially enhances muscle function by accelerating metabolic processes [6] and improving contractile efficiency [7]. However, as exercise continues, muscle temperature can rise beyond the optimal range, potentially leading to performance decrements and premature exercise termination [8]. Intermittent cooling aims to provide a method to attenuate increases in muscle temperature by applying cold water to the limbs distal to the working muscles, potentially reducing excessive heat buildup and thereby enhancing training volume.

The mechanisms behind the potential ergogenic effects of intermittent cooling during resistance exercise are not fully understood, though several theories exist. For example, it is theorized that intermittent cooling enhances performance by affecting central nervous system (CNS) signaling. Research shows that intermittent cooling can increase electromyography (EMG) amplitude, lower the rate of perceived exertion (RPE), and boost perceived arousal during inter-set cooling, possibly improving motor unit recruitment and force production [9,10,11]. Another theory suggests that cooling palms and soles, which contain prominent arteriovenous anastomoses (AVAs), may effectively lower core body temperature. AVAs are responsible for increasing peripheral blood flow to the skin’s surface, leading to increased heat loss and may thus aid in prolonging resistance exercise volume [12,13]. Additionally, intermittent cooling might also reduce the temperature of blood reaching working muscles, preserving the function of temperature-sensitive enzymes like muscle pyruvate kinase [3,14], although current evidence remains inconclusive [15].

Several studies have demonstrated that intermittent cooling can enhance resistance training volume. For instance, palm cooling (PC) or sole cooling (SC) between sets of resistance training has been shown to significantly increase total training volume and muscle activation in upper and lower body exercises in resistance-trained individuals [9,10]. Still, the evidence is not entirely consistent, as other studies failed to observe significant benefits of PC or SC on resistance exercise volume, particularly in untrained individuals during various exercises [3,16,17].

Given the equivocal findings in the literature, the present study aims to provide further insight into the effects of intermittent cooling on resistance exercise volume. For this purpose, we employed three distinct protocols of commonly practiced resistance exercises (pushups, pullups, and leg extensions) in standardized environments to investigate the efficacy of intermittent cooling on training volume in physically active individuals, specifically focusing on the total number of repetitions (TR) performed. Although studies indicated no effect of intermittent cooling on training volume in both untrained and trained individuals, most positive outcomes were observed in participants who were physically active or resistance-trained [9,10,11,14,18]. We therefore hypothesized that intermittent cooling will improve training volume in our sample of physically active individuals. By expanding on existing research, this study aims to further our understanding of how intermittent PC and SC may prolong training volume during resistance exercise.

## 2. Materials and Methods

### 2.1. Ethical Approval

This study was approved by the ethics committee of Leipzig University (ref.-nr. 321/22-ek). In compliance with the Declaration of Helsinki, participants were asked to sign an informed consent form after being briefed about the study goals, procedures, and possible risks.

### 2.2. Experimental Design

This manuscript includes three distinct protocols investigating the effect of intermittent PC or SC on acute resistance exercise performance: Protocol 1 (PULLUP), Protocol 2 (PUSHUP), and Protocol 3 (LEG EXTENSION). All protocols were conducted using a randomized and counterbalanced crossover design. For further details, please see Section 2.4 Procedures.

### 2.3. Participants

Based on previous reports on the effects of intermittent cooling on acute resistance exercise performance, an a priori power analysis was conducted using G*Power version 3.1.9.6. [19]. An average effect size of g_z_ = 0.69 was used for the protocols examining the effects of intermittent PC [10,18], and an average effect size of g_z_ = 0.62 was used for SC [9,11] to determine the required sample size for a one-tailed dependent *t*-test. The sample size required to achieve a power of 80% with a type I error rate of α = 5% was N = 15 for PC and N = 18 for SC.

A total of 15 healthy adults participated in Protocol 1. Three participants had to be excluded from data analysis due to insufficient regeneration between measurements, i.e., by their own account, participants were unable to perform at full physical capacity, resulting in a sample size of 12 participants (age: 25.1 ± 5.5 years, mean ± SD). In Protocol 2, 16 healthy adults were enrolled. Here, 4 participants had to be excluded after the detection of outliers in the total number of repetitions (TR) parameter. Outliers were defined as values exceeding 1.5 times the interquartile range (Q1–Q3) of the sample. The resulting sample size of Protocol 2 was 12 participants (age: 23.5 ± 2.9 years). Finally, 19 healthy adults participated in Protocol 3. Again, as described above, 2 participants had to be excluded due to the detection of outliers, resulting in a final sample size of 17 adults (age: 23.4 ± 3.5 years). All participants had to be between 18 and 35 years of age and were either sports students of Leipzig University or physically active adults with prior experience in unstructured resistance training. Inclusion criteria included the absence of musculoskeletal or neurological disorders and the absence of an acute injury that could impair performance during resistance training. For sample details, please see Table 1.

### 2.4. Procedures

#### 2.4.1. Protocol 1 (PULLUP)

In this randomized crossover experiment, each participant took part in 2 experimental sessions in which 4 sets of bodyweight pullups were performed with 3 min of rest between each set. Intermittent PC was applied during inter-set rest periods in experimental session 1 (PC), while, during experimental session 2, no cooling was applied during inter-set rest periods (NC). The order of PC and NC was randomized and counterbalanced between participants. Experimental sessions were conducted with 7 days of rest in between to avoid impacts of central and peripheral fatigue on resistance exercise performance.

At the beginning of the first experimental session, participants determined their preferred grip width and grip variation, i.e., overhand or underhand, to complete bodyweight pullups on a fixed bar (Barbarian-Line^®^ Profi Half Rack, IFS GmbH, Wassenberg, Germany). These parameters of movement execution were kept constant for both PC and NC. Next, participants performed a general warmup consisting of 5 min of rowing on an ergometer at moderate intensity. Thereafter, a submaximal warmup set of bodyweight pullups was performed. After another 3 min of rest, participants completed 4 sets of pullups with the goal to reach the maximum number of repetitions until muscular failure in each set. Verbal encouragement from the researchers ensured maximum effort. Movement execution during pullups was standardized as follows: Start of the concentric period from a dead-hang until the chin cleared the bar followed by a controlled eccentric period until a dead-hang was reached. Excessive swinging or kicking with the legs to achieve the desired height was prohibited. Furthermore, participants were not allowed to rest longer than 3 s in the starting position of the pullup. Each set was terminated when participants could no longer reach the desired bar height. After each set, participants were seated and either received 3 min of PC (see Section 2.5) or passive recovery (NC). After 3 min of rest, each participant immediately returned to the starting position to perform the next set of pullups. After the completion of the last set, participants were asked to estimate their rate of perceived exertion (RPE) on a visual analog scale (VAS) ranging from 1 (no fatigue) to 10 (maximum fatigue). The number of successful repetitions in each set (SET1–SET4) and the total number of repetitions (TR) performed over 4 consecutive sets of pullups were calculated as outcome parameters of interest.

#### 2.4.2. Protocol 2 (PUSHUP)

Each participant took part in 2 experimental sessions in which 4 sets of bodyweight pushups were performed with 3 min of rest between each set. Intermittent PC was applied during inter-set rest periods in experimental session 1 (PC), while, during experimental session 2, no cooling was applied during inter-set rest periods (NC). The order of PC and NC was randomized and counterbalanced between participants. Experimental sessions were conducted with 7 days of rest in between to avoid impacts of central and peripheral fatigue on resistance exercise performance.

At the beginning of the first experimental session, participants chose their preferred hand placement width during pushup movement execution (wider than shoulder width) which was marked with adhesive tape on a mat placed on the floor. This placement was kept constant during both PC and NC. After that, participants performed a general warmup consisting of mobility exercises of the upper body. Next, a submaximal warmup set of bodyweight pushups was performed. After another 3 min of rest, participants completed 4 sets off pushups with the goal to reach the maximum number of repetitions until muscular failure in each set. Maximum effort was ensured by verbal encouragement. Movement execution of pushups was standardized as follows: Starting position with arms fully extended; tight core, controlled eccentric period until the chest touched the floor; and clasped hands behind the back, concentric period until the arms fully extended. Furthermore, participants were not allowed to rest longer than 3 s in the starting or bottom position of the pushup. Each set was terminated when participants could no longer push up their bodyweight until elbows were fully extended or core tightness was lost. After each set, participants were seated and either received 3 min of PC (see Section 2.5) or passive recovery (NC). After 3 min of rest, each participant immediately returned to the starting position to perform the next set of pushups. After the last set, participants were asked to assess RPE on a visual analog scale (VAS) ranging from 1 (no fatigue) to 10 (maximum fatigue). The number of successful repetitions in each set (SET1–SET4) and the total number of repetitions (TR) performed over 4 consecutive sets of pullups were calculated as outcome parameters of interest.

#### 2.4.3. Protocol 3 (LEG EXTENSION)

This randomized crossover experiment was conducted during a single experimental session. Participants had to perform unilateral leg extensions for 6 consecutive sets with 3 min of rest in between sets on a leg extension machine (Technogym, Cesena, Italy). After completion of 6 total sets with one leg, the contralateral leg performed an additional 6 sets. One of the legs received SC during the 3 min rest periods between successive sets (SC), while the other leg recovered passively (NC). The leg that received SC was randomized at the beginning of the experimental session. Furthermore, the order of SC and NC was randomized and counterbalanced between participants.

At the beginning of this experiment, participants were seated in the leg extension machine which was adjusted to account for individual limb lengths. Participants were then fitted with surface EMG electrodes (see Section 2.6). Thereafter, participants performed a 5 min warmup on a cycle ergometer at low intensity (70–90 Watts). Next, a submaximal warmup set of unilateral leg extension was performed for familiarization. After a 5 min break, participants performed 6 consecutive sets of unilateral leg extensions with a load corresponding to 10% of bodyweight (6.8 ± 1.5 kg) to muscular failure in each set. The contralateral leg rested during the execution of each set. Movement execution was externally cued by a metronome: concentric movement of 1 s duration, isometric contraction for 1 s in the end position of leg extension with fully extended knee joints, eccentric movement of 1 s duration, and 1 s pause in the starting position before starting the next repetition. Movement execution was visually controlled by the researchers and the set was terminated when participants could no longer achieve full knee joint extension or keep the desired pace. Maximum effort was ensured by verbal encouragement of the researchers. Each set was interspaced with a 3 min resting period in which participants remained seated on the leg extension machine. During resting periods, participants either received intermittent sole cooling (SC) or passive recovery (NC). During resting periods, participants were asked to rate their RPE on a visual analog scale (VAS) ranging from 1 (no fatigue) to 10 (maximum fatigue). Tympanic temperature (TT) was assessed by in-ear thermometer (Braun Thermo Scan 7, Kronberg im Thaunus, Germany) before the first set (PRE) and after the completion of each set at the end of the resting period (SET1–SET6). After successfully completing all 6 sets of a condition, participants cycled again for 5 min at low intensity on the cycle ergometer (70–90 watts) and were then allowed to stand up, walk, and rest for 10 min. After this break, participants performed the second series (6 sets) of unilateral leg extension to muscular failure with the contralateral leg. RPE and TT were assessed at identical time-points. The number of successful repetitions in each set (SET1–SET6) and the total number of repetitions (TR) performed over 6 consecutive sets of leg extensions were calculated for each condition (SC and NC) as outcome parameters of interest.

### 2.5. Cooling Device (CoolMitt^®^)

Intermittent PC or SC was applied using a commercially available device (CoolMitt^®^, Arteria Technology, Inc., Orlando, FL, USA). To use this device, hands or feet were placed in heat exchange mitts that were connected to the main CoolMitt unit. In this way, the palms of the hands or soles of the feet were in contact with the heat exchange surfaces in which cool water circulated. The water was circulated by a pump and cooled by a cold-water chamber in which crushed ice was placed. With this device, it is not possible to monitor the exact temperature of the circulating water which ranged from 13 to 15 °C. However, in Protocol 3, the temperature was determined manually using a standard thermometer inserted into the chamber with circulating water (13.4 ± 1.2 °C). Intermittent cooling was applied during the 3 min recovery periods between successive resistance exercise sets in Protocol 1, 2, and 3.

### 2.6. Electromyography (EMG)

In Protocol 3, muscle activity of M. vastus lateralis (VL) and M. vastus medialis (VM) was recorded bilaterally using a wireless Desktop Transmission System (NORAXON Inc., Scottsdale, AZ, USA). Skin preparation, i.e., shaving, abrasion, and cleaning with alcohol was carried out to ensure optimal signal quality during recording. Next, gel-coated self-adhesive surface electrodes (interelectrode distance of 20 mm) were applied on standardized electrode positions according to recommendation by SENIAM [20]. Surface EMG electrodes were pairwise positioned in parallel to the muscle fiber direction. Data were recorded with a sampling rate of 1500 Hz, the input impedance of the amplifier was set at >100 MΩ, bandpass filtering was applied in the frequency range of 10–500 Hz, and common-mode rejection (CMRR) was set at >100 dB.

Furthermore, maximum voluntary contraction (MVC) values were determined for each muscle at the beginning of the experimental day after the general warmup for normalization of EMG activity. Three maximal isometric contractions (5 s) were performed unilaterally in a standardized position using the leg extension machine (knee extension angle 130°) to determine the MVC of VL and VM for each leg separately. Muscle on- and offsets of MVC trials were visually determined by a single trained researcher. In a first step, MVC trials were rectified and mean amplitudes were calculated across each epoch. Amplitude normalization of all leg extension repetitions during SC and NC was carried out using the maximum MVC value of each participant for each muscle separately. A 30 s rest period was granted between each MVC attempt.

Finally, EMG amplitudes of VL and VM were computed from the EMG signal recorded during SC and NC. Therefore, muscle on- and offsets were visually determined by a single trained researcher. Next, mean amplitudes of the rectified signals were calculated. All EMG amplitudes were normalized to individual MVC values. Lastly, MVC normalized EMG amplitudes were averaged for each set (SET1–SET6) and condition (SC, NC) before statistical analysis. Mean amplitude values of 2 participants had to be excluded from the analysis of EMG data due to electrode displacement during recording resulting in a sample size of 15 complete datasets.

### 2.7. Statistical Analysis

All statistical analyses were performed using JASP version 0.18.3 (University of Amsterdam, Amsterdam, The Netherlands). Normality of variables was assessed through Shapiro–Wilk testing (α = 0.05). Sphericity violations of ANOVAS were addressed through Greenhouse-Geisser correction. Effect sizes are reported using η_p_^2^ for ANOVAs, Cohen’s d for paired samples *t*-tests and rank biserial correlation (r_rb_) for Wilcoxon signed-rank tests. Post-hoc analyses of ANOVAs were conducted with Holm correction for multiple comparisons. For all statistical analyses, a *p*-value of *p* < 0.05 was considered significant.

#### 2.7.1. Protocol 1 and 2

The number of repetitions in SET2 to SET4 were normalized to SET1 for each experimental session separately to investigate percentage-wise decreases in performance over the course of one experimental session. Normalized number of repetitions per set and the total number of repetitions (TR) for each condition (PC and NC) were normally distributed. RPE values were not normally distributed.

A repeated-measures ANOVA was used to investigate the impact of intermittent cooling on normalized resistance exercise performance with the within-subject factors CONDITION (PC, NC) and SET (SET1–SET4).

Differences in TR between PC and NC were analyzed using a paired samples *t*-test. In case of RPE, Wilcoxon signed-rank test was utilized.

#### 2.7.2. Protocol 3

The number of repetitions in SET2 to SET6 were normalized to SET1 for each experimental session separately to investigate percentage decreases in performance. Normalized number of repetitions per set and the total number of repetitions (TR) for each condition (SC and NC) were normally distributed. Furthermore, the majority of MVC normalized EMG variables of VL and VM, RPE values, and TT values were normally distributed.

A repeated-measures ANOVA was used to investigate the impact of intermittent cooling on normalized resistance exercise performance with the within-subject factors CONDITION (SC, NC) and SET (SET1–SET6).

Differences in TR between SC and NC were analyzed using a paired samples *t*-test. Changes in TT were assessed through a repeated-measures ANOVA with the within-subject factors CONDITION (SC, NC) and TIME (PRE, SET1–SET6). RPE values were compared between conditions using a repeated-measures ANOVA with the within subject factors CONDITION (SC, NC) and SET (SET1–SET6).

Lastly, MVC-normalized EMG amplitudes were compared between conditions using a repeated-measures ANOVA with the within-subject factors CONDITION (SC, NC), MUSCLE (VL, VM) and SET (SET1–SET6).

## 3. Results

### 3.1. Protocol 1

Repeated-measures ANOVA indicated a significant effect of SET (F_(3, 33)_ = 308.767, *p* < 0.001, η_p_^2^ = 0.966) on normalized pullup performance. Pairwise post hoc comparisons indicated a gradual decrease in repetitions from SET1 to SET2 (mean difference (MD) = −37.9%, standard error (SE) = 2.1%, *p* < 0.001, d = −4.053), SET2 to SET3 (MD = −15.9%, SE = 7.4%, *p* < 0.001, d = −1.699), and SET 3 to SET 4 (MD = −5.1%, SE = 2.1%, *p* = 0.024, d = −0.544). However, no significant effect of CONDITION (F_(1, 11)_ = 0.422, *p* = 0.529, η_p_^2^ = 0.037) or interaction CONDITION × SET (F_(3, 33)_ = 0.777, *p* = 0.515, η_p_^2^ = 0.066) was found (see Figure 1A).

With respect to TR, a significant difference was found with less repetitions performed during PC (32.2 ± 9.5 repetitions) compared to NC (38.2 ± 10.9 repetitions) conditions (MD = −6.0 repetitions, SE = 1.1 repetitions, t_(11)_ = −5.666, *p* < 0.001, d = −1.636, see Figure 1B).

Lastly, no difference in the RPE was found between PC (Median (Mdn) = 7.5, interquartile range (IQR) = 1.0) and NC (Mdn = 7.5, IQR = 2.3) conditions (W = 15.000, z = 0.943, *p* = 0.374, r_rb_ = 0.429).

### 3.2. Protocol 2

Again, repeated-measures ANOVA indicated a significant effect of SET (F_(1_._888, 20_._768)_ = 352.651, *p* < 0.001, η_p_^2^ = 0.970) on normalized pushup performance. Pairwise post hoc comparisons indicated a gradual decrease in repetitions from SET1 to SET2 (MD = −49.6%, SE = 2.3%, *p* < 0.001, d = −5.341) and SET2 to SET3 (MD = −11.6%, SE = 2.3%, *p* < 0.001, d = −1.247). However, no significant effect of CONDITION (F_(1, 11)_ = 0.112, *p* = 0.744, η_p_^2^ = 0.010) or interaction CONDITION × SET (F_(3, 33)_ = 0.397, *p* = 0.756, η_p_^2^ = 0.035) was found (see Figure 2A).

Furthermore, no significant effect of PC (75.0 ± 34.9 repetitions) compared to NC (78.2 ± 31.9 repetitions) was revealed for TR (MD = −3.2 repetitions, SE = 3.2 repetitions, t_(11)_ = −0.988, *p* = 0.344, d = −0.285, see Figure 2B).

Lastly, no difference in the RPE was found between PC (Mdn = 7.5, IQR = 1.0) and NC (Mdn = 7.0, IQR = 1.0) conditions (W = 1.000, z = 1.000, *p* = 1.000, r_rb_ = 1.000).

### 3.3. Protocol 3

Repeated-measures ANOVA indicated a significant effect of SET (F_(2.045, 32.724)_ = 89.444, *p* < 0.001, η_p_^2^ = 0.848) on normalized leg extension performance. Pairwise post hoc comparisons indicated a gradual decrease in repetitions from SET1 to SET2 (MD = −23.1%, SE = 2.3%, *p* < 0.001, d = −1.720), SET2 to SET3 (MD = −8.0%, SE = 2.3%, *p* = 0.004, d = −0.594), and SET3 to SET5 (MD = −8.2%, SE = 2.3%, *p* = 0.003, d = −0.612). However, no significant effect of CONDITION (F_(1, 16)_ = 0.428, *p* = 0.522, η_p_^2^ = 0.026) or interaction CONDITION × SET (F_(5, 80)_ = 1.065, *p* = 0.386, η_p_^2^ = 0.062) was found (see Figure 3A).

TR showed no significant difference between PC (93.6 ± 18.3 repetitions) and NC (93.9 ± 26.5 repetitions) conditions (MD = −0.3 repetitions, SE = 3.3 repetitions, t_(11)_ = −0.090, *p* = 0.929, d = −0.022, see Figure 3B).

Regarding TT, no significant effect of CONDITION (F_(1, 16)_ = 0.545, *p* = 0.471, η_p_^2^ = 0.033), TIME (F_(2.069, 33.107)_ = 0.491, *p* = 0.623, η_p_^2^ = 0.030) or CONDITION × TIME (F_(3.748, 59.963)_ = 1.731, *p* = 0.159, η_p_^2^ = 0.098) was found (see Figure 3C).

Concerning the RPE, a significant effect of SET (F_(2.243, 35.883)_ = 53.881, *p* < 0.001, η_p_^2^ = 0.771) was found. Pairwise post hoc comparisons indicated a gradual increase in the RPE from SET1 to SET2 (MD = 0.5%, SE = 0.2%, *p* = 0.006, d = 0.378), SET2 to SET3 (MD = 0.5%, SE = 0.2%, *p* = 0.004, d = 0.400), SET3 to SET4 (MD = 0.5%, SE = 0.2%, *p* = 0.006, d = 0.378), and SET4 to SET6 (MD = 0.5%, SE = 0.2%, *p* = 0.006, d = 0.378). No significant effect of CONDITION (F_(1, 16)_ = 0.001, *p* = 0.973, η_p_^2^ < 0.001) or interaction CONDITION × SET (F_(2.687, 42.995)_ = 0.374, *p* = 0.750, η_p_^2^ = 0.023) was found (see Figure 3D).

Lastly, MVC-normalized EMG amplitudes showed a significant effect for SET (F_(2.321, 32.493)_ = 34.129, *p* < 0.001, η_p_^2^ = 0.709). Pairwise post hoc comparisons showed a gradual increase in EMG amplitude from SET1 to SET3 (MD = 7.2% MVC, SE = 1.2% MVC, *p* < 0.001, d = 0.256), SET2 to SET4 (MD = 7.7% MVC, SE = 1.2% MVC, *p* < 0.001, d = 0.274), and SET3 to SET5 (MD = 4.5% MVC, SE = 1.2% MVC, *p* = 0.003, d = 0.161). However, no significant effect was found for the factors CONDITION (F_(1, 14)_ = 0.925, *p* = 0.352, η_p_^2^ = 0.062), MUSCLE (F_(1, 14)_ = 0.569, *p* = 0.463, η_p_^2^ = 0.039), CONDITION × SET (F_(2.376, 33.260)_ = 0.119, *p* = 0.917, η_p_^2^ = 0.008) and CONDITION × MUSCLE (F_(1, 14)_ = 1.680, *p* = 0.216, η_p_^2^ = 0.107) (see Figure 3E,F).

## 4. Discussion

The present study aimed to investigate potential ergogenic effects of palm cooling (PC) and sole cooling (SC) on training volume in healthy, physically active participants. Employing three standardized resistance training protocols, we did not observe positive effects of PC or SC on training volume in pullups (Protocol 1), pushups (Protocol 2), or leg extensions (Protocol 3). Notably, we observed that the total number of repetitions (TR) significantly decreased during PC vs. NC for Protocol 1. Moreover, we failed to observe significant differences in tympanic temperature (TT), rate of perceived exertion (RPE), or electromyography (EMG) amplitudes between SC and NC conditions. All findings and their implications are discussed below.

### 4.1. Effects of Intermittent Cooling on Training Volume

The absence of significant differences in training volume between PC and NC conditions in Protocol 1 and Protocol 2 is consistent with certain findings in the literature [3,15,16,17,21], despite some conflicting evidence [9,10,14,18]. Notwithstanding the fundamental heterogeneity pertaining to the ergogenic effects of intermittent cooling, an initial rationale supporting our findings concerns the idea, that the thermoregulatory demands of resistance exercise may not be sufficiently intense to necessitate or benefit from distal cooling interventions. Previous research has established that muscle function is influenced by muscle temperature, with studies indicating that jumping and sprinting performance is compromised at lower muscle temperatures (<36 °C) and enhanced at slightly elevated temperatures (>36 °C). For instance, [22] demonstrated that higher muscle temperatures are associated with improved sprint performance, a finding corroborated by subsequent research involving isokinetic cycling [4]. Crucially, it is still debated, whether or not resistance exercise reliably produces a notable increase in core temperature, specifically in relation to the employed training parameters, i.e., the load, number of sets, and number of repetitions [23]. In Protocol 3, TT measurements, considered to be reliable indicators of core body temperature [24,25], did not reveal differences across six sets of leg extensions, nor between SC and NC conditions, indicating that intermittent cooling did not significantly affect core temperature regulation during leg extensions. On the other hand, skin temperature seems to be affected by cold applications. For example, a study by [26] reported decreased muscle and skin temperatures during cold water immersion after unilateral resistance exercise. Similarly, ref. [27] observed a faster reduction in skin temperature after cold-water immersion following squat exercises, although the application of cold-water immersion did not conclusively alter performance outcomes. Conversely, another study observed relatively stable skin temperatures in the thigh and forearm during a leg press protocol between PC and NC conditions, while only the hand, an area with high AVAs, showed reduced skin temperatures during PC [14]. This suggests that cooling effects on skin temperature are localized and therefore may not significantly impact the broader thermoregulatory processes involved in resistance exercise. An intriguing finding from this study was the significantly greater TR achieved during NC compared to PC in Protocol 1. This effect may be attributed to the nature of the exercise, which involved pullups. The application of PC could have adversely affected grip strength [28] and the tactile sensation of the bar [29], potentially explaining the observed decrease in performance during PC, although further research is necessary to support this assumption. While distal cooling has been shown to reduce core body temperature and enhance performance during aerobic endurance exercises involving significant thermal stress [3,12,13,30,31,32], the relatively minor elevations in core temperature during short, high-intensity resistance exercise sessions, as highlighted by [3] may account, at least in part, for the lack of ergogenic effects observed in our study.

### 4.2. Effects of Intermittent Cooling on Neuromuscular Activity

Concerning the competing idea that intermittent cooling acts through altered CNS signaling, several studies have reported that such applications can lead to increased EMG amplitudes in working muscles, decreased RPE, and heightened levels of perceived arousal compared to NC conditions [9,10,11]. In contrast to the significant increase in EMG amplitude reported by several studies, our findings revealed no differences in EMG amplitudes, or the RPE between SC and NC conditions. Previous studies observed greater EMG activation in working muscles and increased training volume with cooling interventions [9,11], suggesting that cooling may alter muscle function during resistance training. However, our findings align with other studies [16,17], who did not find significant between-group differences in EMG amplitude despite cooling interventions by McMahon et al. [17] specifically highlighted the absence of performance enhancement with PC, suggesting that the shorter cooling duration (1 min) employed in their study could explain the lack of observed benefits. Similarly, Batra et al. [16] reported no differences in EMG activity or training volume during back squats with SC, although the authors did note a reduction in the RPE. It should be emphasized that compared to task-normalized EMG data in studies by Kwon et al. [18] and Cai et al. [9], MVC-normalized EMG data were analyzed in our study and in studies by McMahon et al. [17] and McMahon [33], which represents a further challenge with regard to the comparability of the results. In principle, it can be attested that MVC normalization should be sought in the course of EMG analyses in order to increase the validity of the results.

While multiple studies have reported lower RPE scores with cooling [9,18], our findings do not corroborate this trend. This discrepancy may reflect variations in cooling protocols, participant characteristics, or exercise modalities across studies. Also, it is worth noting that the significant difference in the RPE observed by [18] was only present in one set, while the between-group difference was insignificant. Notably, a follow-up study with all female participants showed no differences in the RPE between PC and NC conditions [10]. Due to the considerable heterogeneity of cooling protocols, it is, however, unfeasible to pin-point the exact reason for the discrepancy of findings in the present literature. However, considering the positive results from endurance protocols, and the link between systemic exertion and core body temperature, it seems reasonable that intermittent cooling only moderates ergogenic effects in situations of extremely high exertion. Consequently, both core body temperature increases and CNS-mediated fatigue perception potentially reach their limits in such situations and, in these cases, may be modulated by intermittent cooling. Resistance exercise protocols such as those used in our study appear to remain unaffected by intermittent cooling.

## 5. Limitations

An initial limitation of this study is the methodological variability between Protocol 1/2 and Protocol 3. Because temperature, RPE, and EMG measurements were not recorded in Protocol 1/2, our analysis was limited to behavioral data. We recognize that in contexts where causality, as in the case of intermittent cooling, remains unclear, including these parameters would provide an additional layer of analysis. This has been addressed in Protocol 3. Future studies should include core temperature, EMG, and RPE measurements without exception to draw conclusions consistent with established explanatory models. In this context, maintaining consistency in the technical setup across participants is crucial when conducting neuromuscular measurements. In Protocol 1, participants were allowed to choose between two grip variations, which likely had a minimal impact on the analyzed outcome measure (TR), but could have introduced variability from a neuromuscular perspective. Furthermore, although the sample sizes are acceptable compared to previous studies, they are still too small to draw definitive conclusions. The heterogeneity in the existing research underscores the need for larger-scale studies. In this context, the variability in study designs must also be considered. While resistance training is a clearly defined term, its practical application can vary greatly. Different objectives such as hypertrophy-, strength-, and endurance-oriented training regimes require different practical approaches, and to fully explore the potential of intermittent cooling, methodological standards must be established and systematically investigated. This is essential to effectively narrow the validity range of intermittent cooling, which in turn would facilitate more precise application recommendations.

## 6. Conclusions

The objective of this study was to evaluate the efficacy of intermittent cooling on training volume during resistance exercise, with a specific focus on the total number of repetitions performed until muscular failure. Our results did not reveal any ergogenic benefits associated with PC and SC during a range of commonly employed resistance training exercises. These findings contribute to the expanding body of literature on the topic, highlighting the variability in the effects of intermittent PC/SC on training volume. Taken together, based on previous studies and the findings from our three protocols, intermittent cooling appears to have limited efficacy as an ergogenic aid. While it presumably poses no harm and may be incorporated into training, current research does not provide sufficient evidence to support its potential for enhancing resistance exercise training volume. Additionally, our findings imply that intermittent cooling might be detrimental to upper body resistance exercise performance where hand grip is a limiting factor (e.g., pullups). Despite the lack of a consistently observable impact of intermittent cooling, future research should adopt a more rigorous approach. This should involve systematically exploring various resistance training protocols under standardized conditions, incorporating physiological measurements of temperature, exertion, and muscle activity to better understand the potential benefits and mechanism of this method in resistance exercise contexts.

## Figures and Tables

**Figure 1 sports-12-00281-f001:**
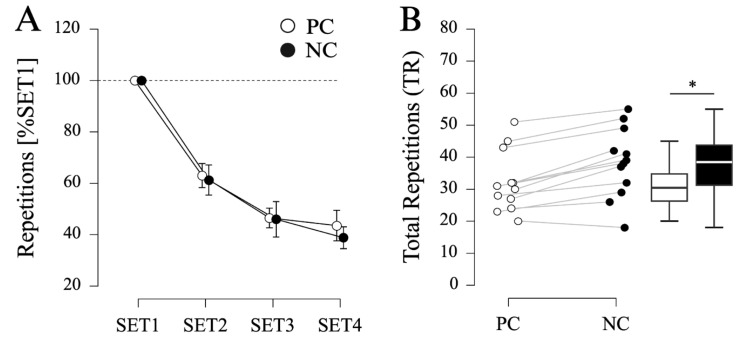
**Results of Protocol 1 (PULLUP).** (**A**) Number of repetitions (normalized to SET1) for palm cooling (PC) and passive recovery (NC) conditions (N = 12). Displayed are mean values, error bars represent 95% confidence intervals. (**B**) Total number of repetitions (SET1–SET4) for PC and NC. Asterisk indicates a significant difference with higher TR in NC compared to PC (*p* < 0.05).

**Figure 2 sports-12-00281-f002:**
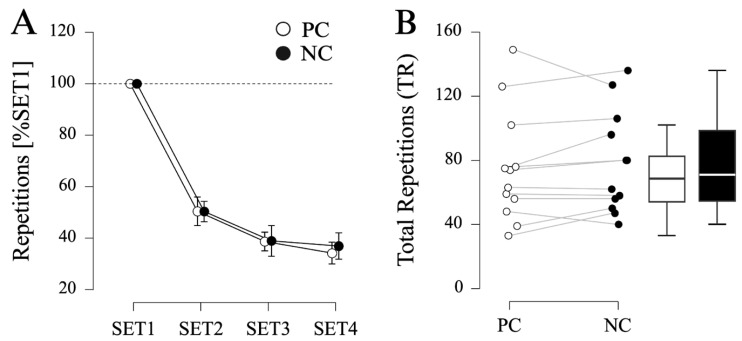
**Results of Protocol 2 (PUSHUP).** (**A**) Number of repetitions (normalized to SET1) for palm cooling (PC) and passive recovery (NC) conditions (N = 12). Displayed are mean values, error bars represent 95% confidence intervals. (**B**) Total number of repetitions (SET1–SET4) for PC and NC.

**Figure 3 sports-12-00281-f003:**
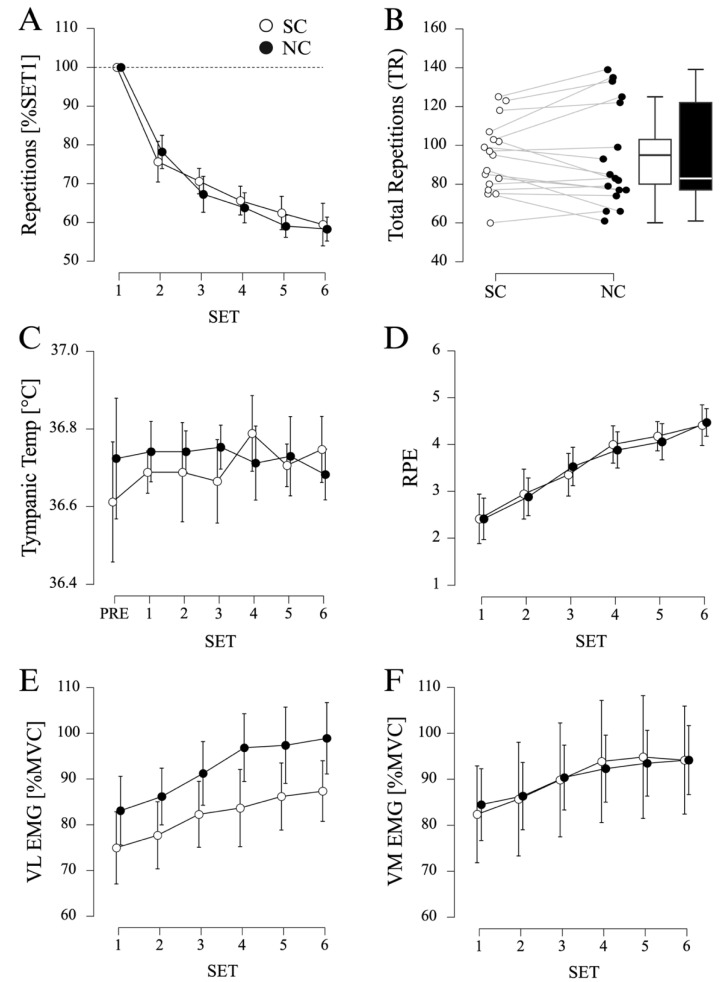
**Results of Protocol 3 (LEGEXT).** (**A**) Number of repetitions (normalized to SET1) for sole cooling (SC) and passive recovery (NC) conditions (N = 17). Displayed are mean values, error bars represent 95% confidence intervals. (**B**) Total number of repetitions (SET1–SET4) for SC and NC (N = 17). (**C**) Tympanic temperature measured via in-ear thermometer (N = 17). (**D**) Rate of perceived exertion (RPE) (N = 17). (**E**,**F**) MVC-normalized mean EMG amplitudes (N = 15) of M. vastus lateralis (VL) and M. vastus medialis (VM).

**Table 1 sports-12-00281-t001:** Anthropometric and demographic data of the protocol samples (values are expressed as mean ± SD).

Variable	Value
**Protocol 1 (PULLUP)**	
Sample size	N = 12 (3 female)
Age (years)	25.1 ± 5.5
**Protocol 2 (PUSHUP)**	
Sample size	N = 12 (2 female)
Age (years)	23.5 ± 2.9
Height (cm)	177.9 ± 8.3
Body mass (kg)	73.8 ± 10.9
Body Mass Index (BMI)	23.2 ± 2.0
Resistance exercise experience (years)	2.7 ± 2.6
Training per week (h)	2.8 ± 2.7
**Protocol 3 (LEG EXTENSION)**	
Sample size	N = 17 (7 female)
Age (years)	23.4 ± 3.5
Height (cm)	171.9 ± 9.7
Body mass (kg)	66.2 ± 12.8
Body Mass Index (BMI)	22.2 ± 2.3
Resistance exercise experience (years)	12.8 ± 7.4
Training per week (hrs)	7.7 ± 3.3

## Data Availability

The original data presented in the study are openly available at https://osf.io/8csdu/ (accessed on 14 October 2024).

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
