# Peer review of "No Effect of Intermittent Palm or Sole Cooling on Acute Training Volume during Resistance Exercise in Physically Active Adults: A Summary of Protocols"

_sports, 2024, doi:10.3390/sports12100281_

Round 1
Reviewer 1 Report
Comments and Suggestions for Authors
Congrats authors for this interesting study. However, I suggest to review the description of the procedure regarding the 3 different protocols: same day, same week,...
How authors control years of experience in pullup?
How authors control training per week in protocols 2 and 3? (I suggest to include also BMI)
Discussion is too dense, please, review and try to avoid abbreviations, it's hard to read, so, to understand. Split the section in subsections.
Attending to this study, practical research, a section of practica applications for researchers and S&C coaches is, almost, mandatory.
Author Response
1) Congrats authors for this interesting study. However, I suggest to review the description of the procedure regarding the 3 different protocols: same day, same week,...
Our response: Thank you for this point. As we stated in the methods section (2.2. Experimental design) all protocols were conducted independently of each other. We further clarified in the abstract, that the 3 protocols investigated were separate experiments, conducted over the course of a year. Each protocol comprised a separate sample, meaning no participant took part in more than one protocol. However, protocols were performed roughly at the same time of day.
2) How authors control years of experience in pullup?
Our response: We did not ask participants to specify their pull-up experience in detail, i.e., years of training or training per week. Therefore, we are unable to control for pullup experience in our main analysis. However, our inclusion criterion for protocol 1 was that participants qualified for participation in this experiment if they could perform at least 5 clean pull-ups in the first set.
3) How authors control training per week in protocols 2 and 3? (I suggest to include also BMI)
Our response: Before the main data analysis we checked if the variable “training per week (h)” had any relationship with the number of pushups/leg extensions achieved in the first set (PC/SC and NC conditions) of protocol 2 & 3. Spearman rank correlation did not reveal any significant associations (all p > 0.05), negating the necessity of including this variable as a covariate. Thank you for your suggestion to include BMI. This variable was added to Table 1.
4) Discussion is too dense, please, review and try to avoid abbreviations, it's hard to read, so, to understand. Split the section in subsections.
Our response: Thank you for this comment. To improve readability, we, firstly, defined all abbreviations used in the discussion at the start of the discussion section. Additionally, the discussion section now contains two subsections to improve clarity.
5) Attending to this study, practical research, a section of practica applications for researchers and S&C coaches is, almost, mandatory.
Our response: Thank you, we agree. The conclusion now includes an additional paragraph concerning practical implications.
“Taken together, based on previous studies and the findings from our three protocols, intermittent cooling appears to have limited efficacy as an ergogenic aid. While it presumably poses no harm and may be incorporated into training, current research does not provide sufficient evidence to support its potential for enhancing resistance exercise training volume. Additionally, our findings imply that intermittent cooling might be detrimental to upper body resistance exercise performance where hand grip is a limiting factor (e.g., pullups).” (lines 501-507)
Reviewer 2 Report
Comments and Suggestions for Authors
The authors have conducted an expansive study. Although not without limitations, I feel with further improvements/ clarifications it would provide a useful addition to cooling literature. As such I have a few queries which I believe may improve the paper and ultimately bring it to a standard for publication.
P1 L 18 please include age, height, mass, sex of participants
P1 L19-21 Abstract results reporting needs significant work. Please include p values alongside description of outcomes. If no main effect of condition is reported please group the effects together e.g. there was no main effect of condition on total reps in the pull-ups, push-ups or knee extensions (all p>0.05)
P1 L23-25 this is fairly redundant information, please use the words available for more detailed results reporting instead.
P1 L28 resistance exercise comes before sole cooling in key words
P1 L34-35 overload is not specific to training volume per se, could this be rephrased?
P1 L38 sustaining suggest maintenance of volume – should this not be increasing, enhancing volume etc?
P2 L69-71 Suggest removing sentence. Grahn et al should not have made this comparison originally as they did not use drugs in their study populations and therefore their comparison lacks any meaningful context.
P2 L73 what does ‘certain conditions’ mean. I would probably emphasize some more of the strengths and weaknesses in study designs of current literature to develop the rationale for this study.
Table 1 – if participants were physically active – why do they have significant resistance training experience? is this unstructured RT experience?
P4 L136 – Why wasn’t pull up variation standardized. The two grip variations result in different biomechanical demands of the latissimus dorsi and also different synergist muscles. This could have potentially added variability to the results
P5 L182-185 why was the leg extensions done in a single session? Although randomized, doing exercise after sets to failure on the contralateral limb could be affected by central command (neural drive), again with the potential to affect results variability
P5 L195 – why use 10% BW for exercise prescription? It makes it difficult to compare against other studies when exercise intensity is unknown
P6 L237 – EMG should be sample at 2K Hz, so 1500 Hz a bit on the low side
P6 L246 was instantaneous peak EMG used or was this averaged over a timeframe?
Results are well reported and visualized.
P11 L376-378 Based on no.of repetitions completed in the results section in push-ups and leg extensions, this would have been mod-heavy exercise prescription. Therefore the authors should include the reference below and add it to the discussion.
McMahon, G., 2024. No Effect of Interset Palm Cooling on Acute Bench Press Performance, Neuromuscular or Metabolic Responses, Following Moderate-Intensity Resistance Exercise. The Journal of Strength & Conditioning Research, 38(7), pp.1213-1220.
P12 L423-434 Its worth highlighting the superior EMG methodologies used in McMahon et al. 2023 and McMahon 2024 as well as those in the current study where normalization was done so using an appropriate MVIC versus task normalization in Kwon and Cai studies.
Comments on the Quality of English Language
Manuscript is largely fine with a few minor improvements
Author Response
The authors have conducted an expansive study. Although not without limitations, I feel with further improvements/ clarifications it would provide a useful addition to cooling literature. As such I have a few queries which I believe may improve the paper and ultimately bring it to a standard for publication.
1) P1 L 18 please include age, height, mass, sex of participants
Our response: This information was added to the abstract.
2) P1 L19-21 Abstract results reporting needs significant work. Please include p values alongside description of outcomes. If no main effect of condition is reported please group the effects together e.g. there was no main effect of condition on total reps in the pull-ups, push-ups or knee extensions (all p>0.05)
Our response: P-values were added to the results reporting.
3) P1 L23-25 this is fairly redundant information, please use the words available for more detailed results reporting instead.
Our response: Thank you for this suggestion. This sentence was removed from the abstract.
4) P1 L28 resistance exercise comes before sole cooling in key words
Our response: Thank you for noticing. We changed the order accordingly.
5) P1 L34-35 overload is not specific to training volume per se, could this be rephrased?
Our response: You are correct, this sentence was not specific enough. We rephrased it as follows:
“Advancing physical performance through resistance training commonly involves progressive overload, a principle, which entails gradually increasing overall training stimulus. This in turn drives continuous adaptations and performance improvement (Kraemer & Ratamess, 2004).” (lines 35-38)
6) P1 L38 sustaining suggest maintenance of volume – should this not be increasing, enhancing volume etc?
Our response: Again, thank you for noticing. We now use “enhancing” instead of “sustaining”.
7) P2 L69-71 Suggest removing sentence. Grahn et al should not have made this comparison originally as they did not use drugs in their study populations and therefore their comparison lacks any meaningful context.
Our response: You are correct, this statement is highly speculative. The sentence was removed.
8) P2 L73 what does ‘certain conditions’ mean. I would probably emphasize some more of the strengths and weaknesses in study designs of current literature to develop the rationale for this study.
Our response: Thank you for noticing. This was not worded correctly. We changed this sentence accordingly.
“Still, the evidence is not entirely consistent, as other studies failed to observe significant benefits of PC or SC on resistance exercise volume, particularly in untrained individuals during various exercises” (lines 72-75)
9) Table 1 – if participants were physically active – why do they have significant resistance training experience? is this unstructured RT experience?
Our response: You are correct. Participants were familiar with unstructured resistance training as part of their university courses or other recreational activities. This was necessary in order to reduce injury risk by ensuring correct movement execution of pullups, pushups or leg extensions under fatiguing conditions. We highlighted this point in the methods section.
“All participants had to be between 18 and 35 years of age and were either sports students of Leipzig University or physically active adults with prior experience in unstructured resistance training.” (lines 119-122)
10) P4 L136 – Why wasn’t pull up variation standardized. The two grip variations result in different biomechanical demands of the latissimus dorsi and also different synergist muscles. This could have potentially added variability to the results
Our response: Thank you for this comment. We did not standardize pull-up grip variations due to the fact that grip variations were held constant on a within-subject level. Additionally, the primary outcome measure for Protocol 1 (pullups) was the number of total repetitions. Naturally, in case of further neuromuscular analyses your point becomes more valid, although neuromuscular analyses were not performed in Protocol 1. Still, we mention this point in the limitation section.
“In this context, maintaining consistency in the technical setup across participants is crucial when conducting neuromuscular measurements. In Protocol 1, participants were allowed to choose between two grip variations, which likely had minimal impact on the analyzed outcome measure (TR), but could have introduced variability from a neuromuscular perspective.” (lines 480-484)
11) P5 L182-185 why was the leg extensions done in a single session? Although randomized, doing exercise after sets to failure on the contralateral limb could be affected by central command (neural drive), again with the potential to affect results variability
Our response: We chose this particular research design (single-session experiment) to minimize the influence of other confounding variables (e.g., diet, sleep quality and quantity, recovery state, stress level) that may occur in multi-day experiments. Indeed, you are right that central fatigue could be a potential confounding factor when both conditions (SC and NC) are performed consecutively in a single session. However, to eliminate this factor, two randomization procedures were performed: 1) participants' legs were randomly assigned to either the SC or NC condition (left or right leg), and 2) the starting leg was randomized independently of randomization 1. In addition, after completing all 6 sets of one leg, a rest period of 20 minutes was inserted before exercising the other leg. With this procedure, we are confident that the potential influence of fatigue on central control does not affect our results.
12) P5 L195 – why use 10% BW for exercise prescription? It makes it difficult to compare against other studies when exercise intensity is unknown
Our response: We see your point. Again, this pertains to the safety aspects mentioned in an earlier response. We intended to refrain from employing maximum force measurements or 1-3 repetition maximum measurements to attenuate potential injury risks of our participants. In future studies enrolling, e.g., elite athletes, maximum force measurements or related assessments should, of course, be utilized to improve comparability between studies.
13) P6 L237 – EMG should be sample at 2K Hz, so 1500 Hz a bit on the low side
Our response: According to the Nyquist theorem, a sampling frequency of at least twice the highest frequency present in the signal is required for EMG recordings. Since the main frequency content of EMG signals falls within the range of 5-500 Hz (Merletti & Di Torino, 1999), a sampling frequency of 1500 Hz is sufficient to analyze EMG signals, particularly for amplitude-related parameters. We are therefore confident in our original analysis.
Merletti, R., & Di Torino, P. (1999). Standards for reporting EMG data. J Electromyogr Kinesiol, 9(1), 3-4.
14) P6 L246 was instantaneous peak EMG used or was this averaged over a timeframe?
Our response: Mean amplitudes of rectified MVC trials (timeframe: EMG burst-onset until burst-offset) were computed. From these mean amplitudes, the highest value was used for MVC-normalization. We clarified this point in the methods section.
“Muscle on- and offsets of MVC trials were visually determined by a single trained researcher. In a first step, MVC trials were rectified and mean amplitudes were calculated across each epoch. Amplitude normalization of all leg extension repetitions during SC and NC was carried out using the maximum MVC value of each participant for each muscle separately.” (lines 253-257)
15) P11 L376-378 Based on no of repetitions completed in the results section in push-ups and leg extensions, this would have been mod-heavy exercise prescription. Therefore the authors should include the reference below and add it to the discussion.
McMahon, G., 2024. No Effect of Interset Palm Cooling on Acute Bench Press Performance, Neuromuscular or Metabolic Responses, Following Moderate-Intensity Resistance Exercise. The Journal of Strength & Conditioning Research, 38(7), pp.1213-1220.
Our response: Thank you for this comment. We agree and now include this reference in the respective section.
16) P12 L423-434 Its worth highlighting the superior EMG methodologies used in McMahon et al. 2023 and McMahon 2024 as well as those in the current study where normalization was done so using an appropriate MVIC versus task normalization in Kwon and Cai studies.
Our response: Thank you for this comment. We have added a paragraph to emphasize this difference.
“It should be emphasized that compared to task-normalized EMG data in studies by Kwon et al. (2010) and Cai et al. (2021), MVC-normalized EMG data were analyzed in our study as well as in studies by McMahon et al. (2023) and McMahon (2024), which represents a further challenge with regard to the comparability of the results. In principle, it can be attested that MVC normalization should be sought in the course of EMG analyses in order to increase the validity of the results.” (lines 450-455)
Round 2
Reviewer 2 Report
Comments and Suggestions for Authors
I'm satisfied with the authors' responses and congratulate them on their study.